# High molecular weight glutenin gene diversity in *Aegilops tauschii* demonstrates unique origin of superior wheat quality

Emily Delorean[1,23], Liangliang Gao[1,23], Jose Fausto Cervantes Lopez[2], Open Wild Wheat Consortium*, Brande B. H. Wulff [3], Maria Itria Ibba [2] & Jesse Poland [1,3✉]

Central to the diversity of wheat products was the origin of hexaploid bread wheat, which added the D-genome of *Aegilops tauschii* to tetraploid wheat giving rise to superior dough properties in leavened breads. The polyploidization, however, imposed a genetic bottleneck, with only limited diversity introduced in the wheat D-subgenome. To understand genetic variants for quality, we sequenced 273 accessions spanning the known diversity of *Ae. tauschii*. We discovered 45 haplotypes in *Glu-D1*, a major determinant of quality, relative to the two predominant haplotypes in wheat. The wheat allele *2 + 12* was found in *Ae. tauschii* Lineage 2, the donor of the wheat D-subgenome. Conversely, the superior quality wheat allele *5 + 10* allele originated in Lineage 3, a recently characterized lineage of *Ae. tauschii*, showing a unique origin of this important allele. These two wheat alleles were also quite similar relative to the total observed molecular diversity in *Ae. tauschii* at *Glu-D1*. *Ae. tauschii* is thus a reservoir for unique *Glu-D1* alleles and provides the genomic resource to begin utilizing new alleles for end-use quality improvement in wheat breeding programs.

[1] Department of Plant Pathology, Kansas State University, Manhattan, KS, USA. [2] Global Wheat Program, International Maize and Wheat Improvement Center (CIMMYT), Mexico, DF, Mexico. [3] King Abdullah University of Science and Technology, Thuwal, Saudi Arabia. [23]These authors contributed equally: Emily Delorean, Liangliang Gao. *A list of authors and their affiliations appears at the end of the paper. ✉email: jpoland@ksu.edu

Originating in the Fertile Crescent some 10,000 years ago, hexaploid wheat (*Triticum aestivum*) is now grown and consumed around the world[1]. The global consumption of wheat as a staple crop is owed principally to the unique viscoelastic properties of wheat dough that lend it the capacity to make diverse baked products such as leavened bread, tortillas, chapati, pastries, and noodles. The uniqueness of wheat dough can also be described as the strength to resist deformation and elasticity to recover the original shape as well as the viscosity to permanently deform under persistent stress. Elasticity is important for the product to hold shape, while viscosity allows the dough to be worked and formed. The balance of the competing properties determines what baked goods a dough is suitable for, such as a dough with greater strength for leavened pan bread compared to the more extensible dough that is desired for a chapati or tortilla.

Bread wheat is an allohexaploid with the A-, B- and D-subgenomes contributed by different, but related, species. The closest relative to the wheat A-subgenome is diploid *Triticum urartu*, with other diploid A-genome species including the wild and domesticated Einkorn wheat (*Triticum monococcum*). While the exact ancestor of the B-genome is unknown and presumed extinct, it is believed that *Ae. speltoides* (S-genome) is the closest living relative. These two species were brought together to form a tetraploid wheat species with AABB genome composition, which is known as durum or pasta wheat (*Triticum durum*). The D genome from *Aegilops tauschii* was the most recent addition forming a hexaploid wheat species. This addition of the D-subgenome led to a much broader adaptation and superior bread-making quality compared to the tetraploid and diploid ancestors[2]. However, the original hexaploid species originated from very few *Ae. tauschii* accessions and limited subsequent cross-hybridization likely caused by ploidy barriers with the diploid *Ae. tauschii*[3]. This genetic bottleneck resulted in limited genetic diversity in the wheat D-subgenome[4].

*Aegilops tauschii* as a species is comprised of three subspecies, commonly referred to as lineages, that can be discerned through subtle phenotypic differences but more readily differentiated genetically by large variation among the lineages[5,6]. Lineage 1, *Ae. tauschii* spp. *tauschii*, is found across the native range of *Ae. tauschii* while Lineage 2, *Ae. tauschii* spp. *strangulata* is found primarily near the Caspian Sea in Iran and Azerbaijan[5]. Recent work has confirmed the existence of another lineage, Lineage 3, which is not yet designated by one of the historical subspecies names. While Lineage 2 is known to be a wheat D-subgenome donor, recent work has shown that Lineage 3 also contributed to the modern wheat genome[6].

The utility of wheat, the variation of wheat products, and therefore consumption is driven by the strength and elasticity of the dough which is determined by the structure of the gluten matrix. This matrix is formed from a combination of high-molecular weight (HMW) and low-molecular weight (LMW) glutenin proteins and gliadins[7]. The backbone of the gluten matrix is developed during dough mixing by the covalent disulfide bonds between cysteine residues in HMW glutenins[8]. These glutenins, therefore, are some of the most important genes giving wheat its unique dough properties. They are encoded by a relatively simple locus on the long arm of the group one chromosomes of the Triticeae. Hexaploid wheat, comprised of the A-, B- and D-genomes, thus contains three HMW glutenin loci; *Glu-A1*, *Glu-B1* and *Glu-D1*, respectively. Each locus harbors two HMW glutenin genes known as the *x* and *y* subunit, which are tightly linked but separated by tens to hundreds of kilobase pairs (kb)[9–11]. Each subunit consists of short, unique N and C terminal domains which flank a highly repetitive central region that accounts for 74–84% of the total protein length[12].

Allelic differences in all three glutenin loci contribute to the conformation of the gluten matrix and variability in end-use quality. The D-subgenome locus, however, is the major driver of bread quality and the absence of the D-genome leads to substantially different dough qualities found in tetraploid pasta wheats[13]. The two common alleles at *Glu-D1* found in bread wheat are *Glu-D1a* (SDS-PAGE allele designation *2 + 12*) and *Glu-D1d* (*5 + 10*), with the latter being associated with superior bread-making quality[14–17]. Following the domestication and breeding of wheat, there is limited variation at the *Glu-D1* locus in the D-genome with only these two alleles found in the vast majority of bread wheat throughout the world[18,19]. Thus, the addition of the wheat D-subgenome and specifically variation at *Glu-D1* has a substantial impact on wheat quality globally. This is arguably the single greatest defining feature of bread wheat.

Reflecting the importance of *Glu-D1* in determining the end-use quality of wheat, focus has been given to understanding the variation present in *Ae. tauschii* for this locus. Much of the work has utilized sodium dodecyl sulfate-polyacrylamide gel electrophoresis (SDS-PAGE) protein analysis of *Ae. tauschii* collections[20–23]. From this work, over 37 SDS-PAGE *Glu-D1* alleles have been named in *Ae. tauschii*. However, due to the limited resolution of SDS-PAGE, many of the alleles have indistinguishable SDS-PAGE mobilities from the common *Glu-D1* hexaploid alleles, *2 + 12* and *5 + 10*, or are difficult to reliably distinguish. By changing the polyacrylamide percentage or acidity in the SDS-PAGE, it was shown that the *Ae. tauschii 2 + 12* and *5 + 10* alleles were slightly different than the common wheat alleles[24]. These *Ae. tauschii* alleles are therefore given the designations *2t + 12t* and *5t + 10t*. In addition to the *2t + 12t* and *5t + 10t* alleles, a large number of SDS-PAGE alleles have been described, supporting the hypothesis that *Ae. tauschii* could be a vast resource for untapped diversity at *Glu-D1* and that this diversity could be utilized for wheat quality improvement.

Here we characterized the *Glu-D1* allelic diversity in a panel of 273 sequenced *Ae. tauschii* accessions. The panel spans the known genetic diversity of *Ae. tauschii* and is a powerful resource for association mapping and gene identification[6]. From the sequenced *Ae. tauschii* panel, we discovered hundreds of genetic variants which defined dozens of unique haplotypes at *Glu-D1*. This gives the needed molecular information to track these alleles in breeding germplasm, which will, in turn, enable targeted assessment of the *Ae. tauschii* HMW glutenin alleles in hexaploid backgrounds leading to utilization of favorable alleles for wheat quality improvement.

## Results and discussion

**Molecular diversity of Glu-D1 in *Ae. tauschii*.** Through the Open Wild Wheat Consortium, we obtained Illumina 150 bp paired-end short reads from 273 *Ae. tauschii* accessions, of which 234 were unique, each sequenced to greater than 7-fold coverage[6]. These were aligned to the *Ae. tauschii* 'AL8/78' reference genome and sequence variants at the annotated *Glu-D1* locus were extracted for *Glu-D1x* at chr1D: 419306988-419309556 and *Glu-D1y* at chr1D: 419364015 -419365995 (Supplementary Data 2). We also included wheat cultivars in this analysis to compare *Ae. tauschii* variants to the common hexaploid wheat alleles *5 + 10* (varieties 'CDC Stanley' and 'CDC Landmark') and *2 + 12* (varieties 'Chinese Spring' and 'LongReach Lancer'). From this panel, we identified a total of 308 variants at *Glu-D1*, which were used to generate haplotypes and evaluate molecular diversity at this locus.

From the *Ae. tauschii* germplasm collection we identified 32 and 33 haplotypes within the coding sequence for the *x* and *y* subunits of the *Glu-D1* locus, respectively (Supplemental Data 1). When considering the complete *Glu-D1* locus with a combination of the *x* and *y* subunit, a total of 45 haplotypes were identified.

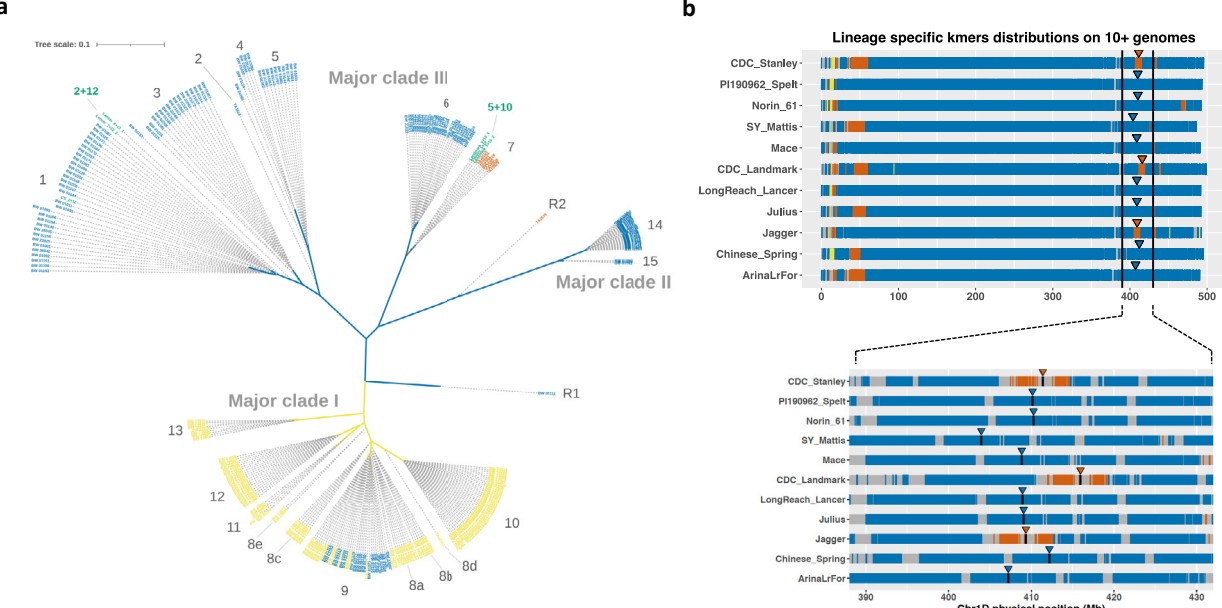

**Fig. 1 Phylogenetic tree of *Aegilops tauschii* and bread wheat accessions based on *Glu-D1* haplotypes and origin of wheat *5 + 10* allele from Lineage 3. a** Maximum likelihood tree based on 308 single-nucleotide variants found in *Glu-D1 x* and *y* subunits including 2.5 kb flanking regions on each side of the genes. The corresponding lineage of each accession is colored in yellow (Lineage 1), blue (Lineage 2), orange (Lineage 3), and green (bread wheat). Haplotypes with major-clades and subclades are designated by numbers. The wheat *2 + 12* allele in subclade 1 and the *5 + 10* allele is found in subclade 7. (https://itol.embl.de/export/17410641549711623435886). R1 and R2 designate putative recombinant Glu-D1 haplotypes in Ae. tauschii accessions TA1668 (R1) and TA2576 (R2). **b** Lineage-specific *k*-mers bins (100 Kb) as identified by Gaurav et al. anchored on genome assemblies of 11 wheat cultivar shown for Lineage 1 (yellow), Lineage 2 (blue), and Lineage 3 (orange). Arrows and black segments indicate the positions of *Glu-D1* in each of the bread wheat genomes with *2 + 12* in blue and *5 + 10* in orange.

The various *x* and *y* subunit haplotypes were almost exclusively associated with each other, demonstrating the close physical association and limited recombination between the two genes. We included the 2500 bp up- and down-stream sequences of each subunit in our analysis to see if this resulted in further differentiation of alleles as short-read sequences often do not align uniquely to the central, highly repetitive region of the HMW glutenin genes. Including the flanking regions did not result in additional haplotypes. Thus, it appears that the identified variants are sufficient for reliably differentiating alleles at *Glu-D1*.

Gene-level phylogeny at *Glu-D1* for all of the *Ae. tauschii* accessions revealed that haplotypes clustered into three major clades, two of which were associated predominantly with Lineage 2 and one with Lineage 1 (Fig. 1a). A unique group of *Glu-D1* alleles from the newly characterized Lineage 3 accessions were found within a narrow clade with Lineage 2. Among the three major clades, we designated 15 subclades. Of the 15 subclades, eight were associated exclusively with Lineage 2, five with Lineage 1, and one with Lineage 3. The Lineage 3 accessions all fell within the Lineage 2 major-clades but occupied a unique subclade therein. Thus, the gene-level phylogeny at this locus agrees very closely with the previously described population structure of the *Ae. tauschii* lineages[5,6]. We also observed one clade (9) that had representative accessions from both Lineage 1 and 2. This could represent an ancestral haplotype found in both lineages which underwent incomplete lineage sorting, or a case of recent interlineage haplotype exchange. Cases of haplotypes shared across Lineages 1 and 2 were also observed for pest (*Cmc4*) and disease resistance (*Sr46*) genes[6].

Lineage 2, the recognized ancestral diploid donor of the D-subgenome of hexaploid wheat[3], had greater *Glu-D1* molecular haplotype diversity than Lineage 1. Not only were there more subclades associated with Lineage 2, there were also more

haplotypes. As expected, the haplotypes of wheat clustered within Lineage 2 subclades (Fig. 1). Within Lineage 2, we observed *Ae. tauschii* accessions with a matching sequence haplotype to the wheat *2 + 12* allele consistent with the D-subgenome origin from Lineage 2. Interestingly, the wheat *5 + 10* allele clustered within the Lineage 3 subclade indicating a unique origin of this allele. We then examined lineage-specific *k*-mer bin distributions as determined by Gaurav et al.[6] and found that Lineage 3-specific *k*-mers co-colocalized with bread wheat *5 + 10* allele in the genomes of a set of 11 bread wheat cultivars (Fig. 1b), giving additional support that the superior *5 + 10* allele for bread wheat quality was derived from Lineage 3. These findings reveal that the Lineage 3 contribution to the wheat D-subgenome included the very valuable *Glu-D1 5 + 10* allele, arguably one of the most important factors defining the quality of bread wheat.

Given the large difference in quality between wheat cultivars carrying *2 + 12* and *5 + 10* alleles, we hypothesized that these two haplotypes would not be similar at a molecular level. However, we found that *2 + 12* and *5 + 10* clustered relatively closely within major-clade III, with much greater overall diversity detected across *Ae. tauschii* particularly when including the Lineage 1 accessions which had very different haplotypes. When comparing the *2 + 12* and *5 + 10* haplotypes to those found in Lineage 1, it becomes apparent that *Ae. tauschii* carries alleles that are very unlike anything seen in bread wheat and may offer unique functional characteristics when introgressed into hexaploid backgrounds.

**Geographic diversity.** Given the known geographic structure and distribution of *Ae. tauschii* which is associated with various levels of population structure[5], we evaluated the *Glu-D1* diversity relative to the geographic origin of the *Ae. tauschii* accessions.

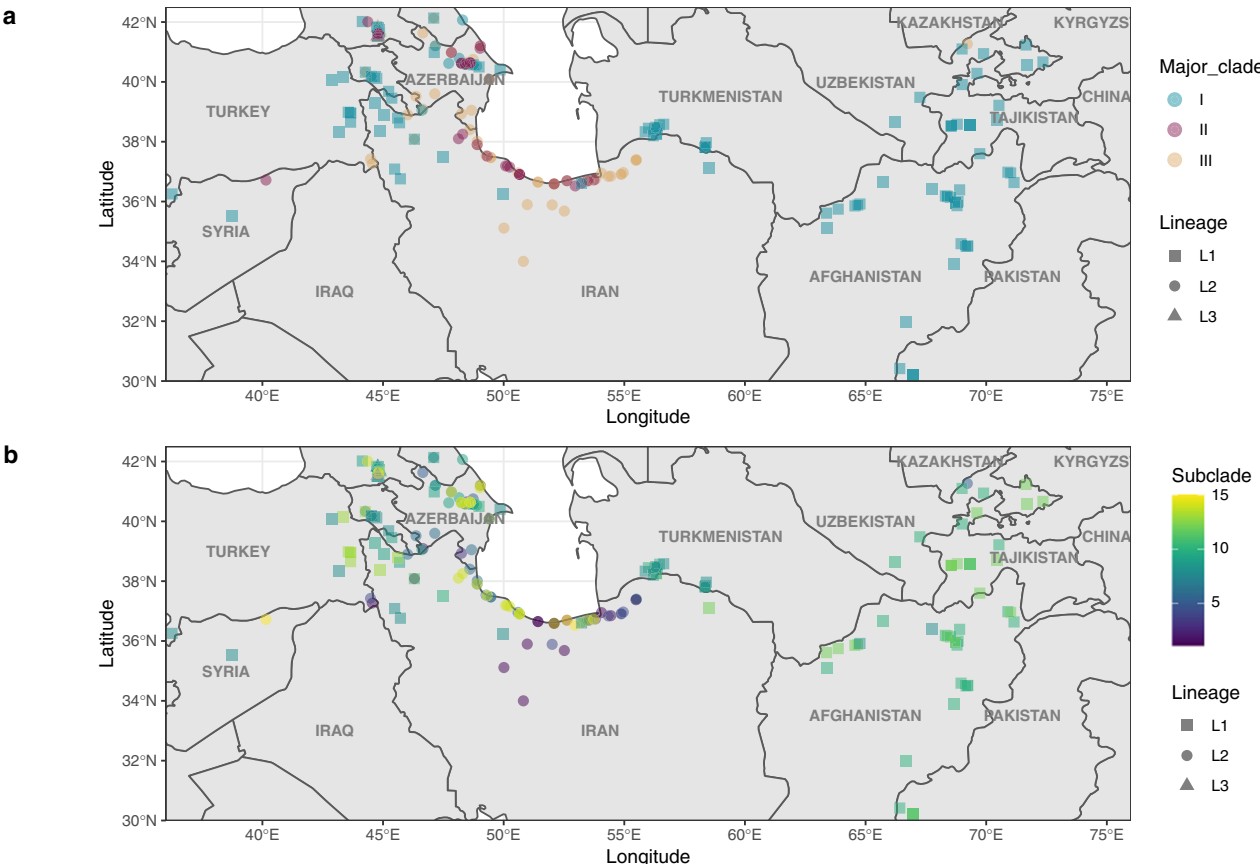

**Fig. 2 Geographic distribution of Glu-D1 haplotypes.** Molecular haplotypes for *Glu-D1* shown at the collection site for the given Ae. tauschii accession. **a** Distribution of accessions according to *Glu-D1* major-clades with major-clade I in blue, major-clade II in red and major-clade III in orange. **b** Distribution of accessions according to Glu-D1 haplotype subclades. Haplotypes are shown on a scale from purple to yellow according to the phylogenetic tree (Fig. 1). Lineages are shown as squares for Lineage 1, circles for Lineage 2 and triangles for Lineage 3.

*Glu-D1*major-clades II and III overlapped in regions around the Caspian Sea whereas major-Clade I spanned the entire geographic range (Fig. 2a). Molecular haplotypes were strongly associated with geographic origin, consistent with the overall genome-wide picture[5], and genetic distances between alleles increased with the geographic distance between collection sites of the *Ae. tauschii* accessions (Fig. 2b). The greatest concentration of haplotype diversity was located along the shores of the Caspian Sea in Iran (Fig. 2b). Consistent with a hypothesis of admixture between Lineage 1 and Lineage 2 leading to shared gene-level haplotypes across the lineages, the accessions from Lineage 1 and 2 with the same *Glu-D1* haplotype (within subclade 9) were collected very near one another.

**Molecular haplotypes identify novel Glu-D1 alleles**. We employed SDS-PAGE analysis, the traditional standard for differentiating HMW glutenin alleles, to determine if the haplotype molecular sequence diversity would also reflect differences in protein mobility. We evaluated a total of 72 unique accessions with SDS-PAGE and differentiated 9 alleles for the *x* subunit and 8 alleles at the *y* subunit from this protein mobility assay. Analysis of the Lineage 1 and Lineage 2 variants revealed that molecular haplotypes were consistent with the proteins differentiated by SDS-PAGE (Supplemental Data 1). For the majority of the alleles that were differentiated by SDS-PAGE, we were able to unambiguously correlate the observed SDS-PAGE alleles with the molecular variants. Although specific molecular haplotypes were

associated with specific SDS-PAGE mobilities, there was little concordance between gene-level variation and SDS-PAGE mobility as similar alleles at the molecular level were observed with very different SDS-PAGE mobilities. Alternatively, very different molecular haplotypes were observed with the same SDS-PAGE allele. This supports our hypothesis that the observed sequence variants are effectively in complete linkage disequilibrium and tagging the variation from the central repeat region. Similarly, the SDS-PAGE diversity was lower than that of the molecular haplotypes, thus SDS-PAGE has less power to differentiate alleles than the molecular haplotypes. As noted, the same SDS-PAGE mobilities were observed in both Lineage 1 and Lineage 2 haplotypes, but the molecular haplotypes were clearly differentiated (Fig. 1a and Supplemental Data 1). The protein mobility differences are considered to be primarily due to variation in the central repetitive region and therefore are not directly detectable with short-read sequencing, though the variable central repeats are completely phased with diagnostic haplotype variants within the terminal coding regions. Thus, we conclude that a sequence-based resource such as this *Ae. tauschii* panel provides a superior tool for the identification and tracking of unique *Glu-D1* alleles in molecular breeding.

We also examined the connection between the glutenin protein mobility in *Ae. tauschii* compared to hexaploid wheat. *Ae. tauschii* haplotype *Dx1a + Dy1a* matched with the wheat *2 + 12* haplotype and exhibited the same SDS-PAGE mobility. Although we found an *Ae. tauschii* haplotype identical to the wheat *2 + 12* allele haplotype, the exact wheat *5 + 10* haplotype was not

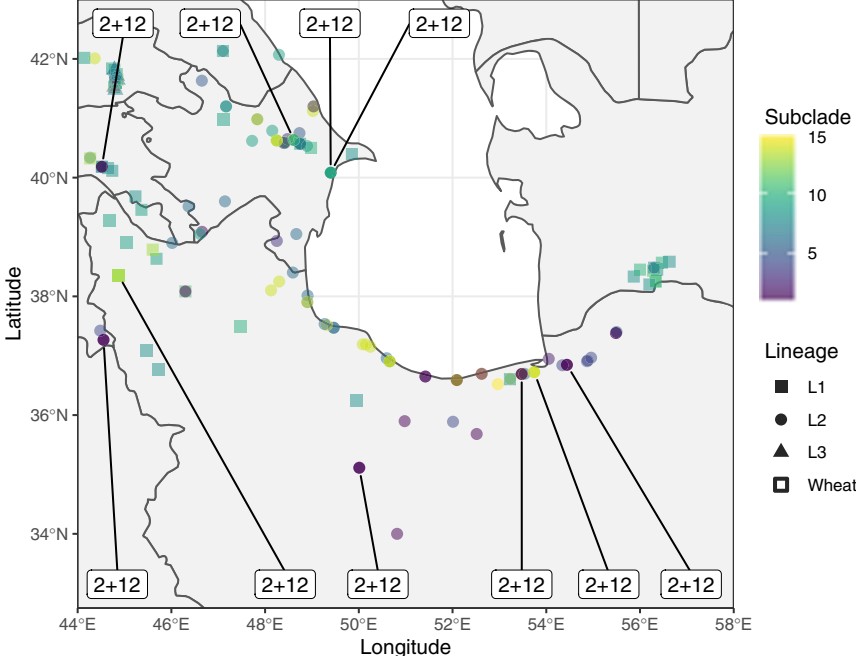

**Fig. 3 Cryptic haplotypes within SDS-PAGE alleles.** The molecular haplotypes for *Ae. tauschii* accessions at the sites where the accessions were collected. Corresponding SDS-PAGE allele is noted for *2 + 12* mobility alleles.

detected in this panel, however, a very closely related Lineage 3 haplotype was found. Additionally, no *5 + 10* SDS-PAGE mobilities were observed. This was a surprising observation given that previous studies reported *Ae. tauschii* alleles with a *5 + 10* SDS-PAGE mobility[20]. However, Williams et al.[20] did not report the identities of the *Ae. tauschii* accessions with *5 + 10* SDS-PAGE mobility. The haplotype *Dx7a + Dy7a* in the newly characterized Lineage 3[6] was most similar to *5 + 10* on the molecular level, however, it carried eight variant differences. This current panel, however, only has five unique accessions representing Lineage 3. It is likely therefore that exploration of additional Lineage 3 accessions could reveal a haplotype exactly matching the wheat *5 + 10* with the same mobility.

**Cryptic haplotypes.** One of the most valuable findings of this study was the high prevalence of cryptic molecular haplotypes hidden within SDS-PAGE mobilities. Within every SDS-PAGE mobility pattern, there were multiple molecular haplotypes, often from very different subclades and occasionally from entirely different major clades. The cryptic SDS-PAGE haplotypes, accordingly, were geographically disperse (Fig. 3). For example, within SDS-PAGE *2 + 12* were four haplotypes; one which was the same as wheat *2 + 12* (*Dx1a + Dy1a*), another which was within the same subclade (*Dx1c + Dy1d*), and two from entirely different major-clades (*Dx9a + Dy9b* and *Dx13b + Dy13a*). Also, within subclade 9 were the SDS-PAGE mobilities *Dx2 + Dy10* and *Dx2 + Dy11*, and within subclade 13 were the SDS-PAGE mobilities *1t + 12*, *2.1\* + 12.1\**, and *4 + 10* further supporting that these haplotypes are not all similar to the wheat *2 + 12* haplotype at the molecular level. However, the proteins still migrate similarly on an SDS-PAGE. These results suggest that SDS-PAGE alone is insufficient when characterizing HMW glutenin diversity in wild relatives and will not be a suitable tool for tracking unique alleles in the hexaploid wheat germplasm.

While most molecular haplotypes delineated along the three *Ae. tauschii* lineages (Fig. 1), a notable exception was within the predominantly Lineage 1 major-clade, subclade 9, where the same three haplotypes (*Dx9a + Dy9a*, *Dx9a + Dy9b*, and *Dx9a*

*+ Dy9c*) were observed in both Lineage 1 and Lineage 2 accessions. Interestingly, while there were three haplotypes at the *y* subunit, there was only a single *x* haplotype associated with all three of these. The *x* subunit mobility was the same for all three haplotypes, indicating that the *x* allele is in fact the same. However, the *y* subunit was differentiated with the mobility *Dy9b* was faster than that of *Dy9a* and *Dy10c* (Supplemental Fig. 1).

**Recombinant haplotypes identified.** The close proximity of the glutenin genes results in such tight linkage that recombination is extremely rare. To date, a recombination between the *x* and *y* subunit of any HMW-GS locus has yet to be verified. Among the 242 *Ae. tauschii* accessions studied here, we found a clear example of a historical recombination at *Glu-D1* in the Lineage 2 accession TA1668 (BW_01111). SDS-PAGE mobility of TA1668 matches that of TA10081 (BW_01039), *Dx2 + Dy10.2*. TA1668 and TA10081 share the same *y* subunit haplotype, *Dy5a* of major-clade III. However, the *x* subunit of TA1668 corresponds to haplotype *Dx9a* of major-clade I (Fig. 4a). The corresponding subclade 9 is the only subclade found in both Lineage 1 and Lineage 2 accessions, indicating that there was incomplete lineage sorting or admixture between the two lineages that lead to the introgression of a Lineage 1 *Glu-D1* haplotype into the Lineage 2 population. In the overlapping geographic range of both haplotypes, it appears there was a rare recombination between the Lineage 1 and Lineage 2 *Glu-D1* alleles, leading to the recombinant *Dx9a + Dy5a* found in TA1688 (Fig. 4c).

The Lineage 3 accession TA2576 also appears to carry a recombinant haplotype (*Dx7b + Dy15b*) (Fig. 4b). However, our data set did not contain the exact haplotypes involved in the recombination that led to *Dx7b + Dy15b*. The closest *x* subunit haplotype is *Dx7a*, the only other Lineage 3 haplotype, from major-clade III and the closest *y* subunit is the Lineage 2 haplotype *Dy15a* from major-clade II (Lineage 2). We, therefore, designated the *x* and *y* subunit haplotypes of TA2576 haplotypes within subclades 7 and 15. Geographical analysis reveals that TA2576 was collected from a region shared with other Lineage 3 accessions (Fig. 4d). However, the accessions containing *Dy15a*

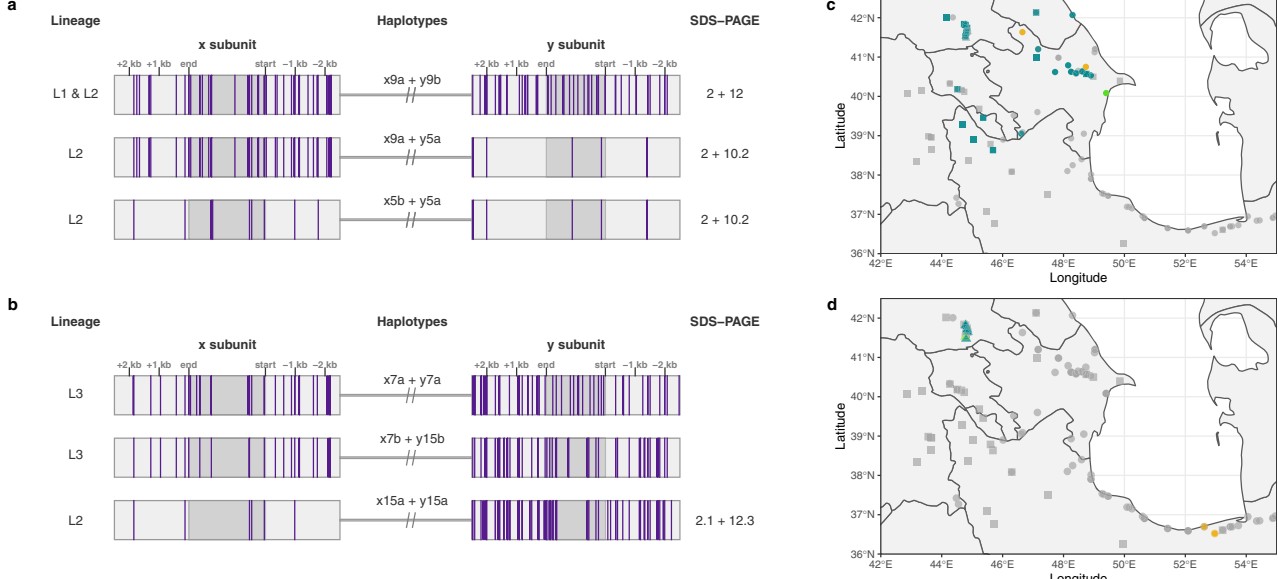

**Fig. 4 *Glu-D1* recombinants.** Molecular haplotype representation of recombinant *Glu-D1* haplotypes for accessions **a** TA1668 (Lineage 2) and **b** TA2576 (Lineage 3). Vertical purple bars represent the alternate allele variants as called against the AL8 7/8 genome assembly. SDS-PAGE allele for the given haplotypes are shown to the right of each gene model. Haplotype *Dx9a* is present in both Lineage 1 and Lineage 2 accessions. The closest potential *x* and *y* subunit haplotypes involved in the recombinant haplotype *x7b + y15b* of TA2576 are *Dx7a* (Lineage 3) and *Dy15a* (Lineage 2). SDS-PAGE protein mobilities for *Dx7a + Dy7a* and *x7b + y15b* were not analyzed. Geographical distribution of recombinant *Glu-D1* haplotypes for **c** TA1668 (BW_01111), Lineage 2, and **d** TA2576, Lineage 3. Collection site of recombinant accessions is marked in lime green, whereas turquoise and orange designate the collection sites of accessions carrying *x* subunit and *y* subunit haplotypes, respectively. Accessions with unrelated haplotypes are in light gray. Lineages are shown in squares (Lineage 1), circles (Lineage 2), or triangles (Lineage 3).

haplotype were not collected from a shared region with the Lineage 3 accessions. Although not conclusive, the most parsimonious explanation is therefore that $Dx7b + Dy15b$ represents a recombinant haplotype between the *x* and *y* subunits from two different alleles. Within our current panel, however, we are unable to differentiate exactly which original haplotypes gave rise to this recombinant haplotype.

**Functional variation in Glu-D1.** To see if any of the variants with the coding region were potentially functional, we compared deduced amino acid sequences based on the reference genome and single nucleotide variant information. We were specifically looking for changes in cysteine residues and other non-synonymous mutations leading to the amino acid changes, but we did not detect any changes in cysteine residue composition in the N or C terminal regions (Supplemental Data 3). However, our results do show that a majority of variant sites resulted in non-synonymous changes in amino acid compositions (e.g., 39 out of 68 in *Glu-D1y*) supporting a hypothesis that these storage proteins are not under a purifying selection and any of these mutations could have a potentially substantial impact on quality (Supplemental Data 3). Nonetheless, the superior $5 + 10$ allele again clustered very tightly with Lineage 3 accessions based on the protein sequence (Supplemental Figs. 2 and 3).

The *Glu-D1* locus of wheat provides the greatest contribution to gluten strength, regardless of the allele present[13]. The allelic diversity of *Glu-D1* in wheat is limited to two predominant alleles $2 + 12$ and $5 + 10$, and a few rare alleles (i.e., $3 + 12$, $4 + 10$) which are associated with similar end-use quality as $2 + 12$[25,26]. The unique $2.2 + 12$ SDS-PAGE allele, which is found at high frequency in Japanese wheat, was shown to be identical to the $2 + 12$ haplotype with the exception of additional repeats in the internal repeat domain of the *x* subunit[25,27,28]. The *x* subunit protein from $5 + 10$ has a unique cysteine residue just within the central repeat

domain which is suspected to increase disulfide bonds in the forming dough. The early expression and transcription of this allele are also higher than that of the other *Glu-D1* alleles, in particular $2 + 12$[16]. Other factors underlying the functionality of high-molecular weight glutenin alleles in quality have also been proposed, such as the length and composition of the central repeat domain[29,30], the position and number of cysteine residues for disulfide bonds[31], the glutamine composition for hydrogen bonds[31,32], and promoter elements[33,34]. At this time, it is unclear which of these characteristics, or the combination and interaction of different features, lend to superior quality characteristics and laboratory testing of end-use quality is necessary to determine superior and inferior alleles. As the first step needed to identify superior quality alleles, we developed *Ae. tauschii Glu-D1* haplotype groups from dense molecular markers. This haplotype information will help breeders target novel alleles not currently found in the hexaploid wheat germplasm for advancement to quality testing stages. While much work remains to be done on the functional variants within *Glu-D1* and other glutenin genes, the determination of novel variants and unique haplotypes gives priority for the extensive germplasm development and testing. In the absence of functional variant knowledge, the assessment of molecular diversity in this study, however, supports the approach of identifying novel haplotypes followed by targeted testing in the breeding pipelines as the best path forward for wheat improvement. The unique origin of $5 + 10$ from Lineage 3, however, further supports the important contributions of genetic diversity from *Ae. tauschii* to the wheat D genome consistent with the findings by Gaurav et al.[6].

**Conclusions**
Our haplotype analysis revealed that the *x* and *y* subunits are strongly associated even in diverse germplasm and that the *Glu-D1* haplotypes were clustered to specific geographic origins. Consistent with the findings of Gaurav et al.[6], we found evidence of two lineages (Lineage

2 and Lineage 3) contributing to the D genome of wheat with the superior $5 + 10$ allele found associated with Lineage 3 accessions. Given the excellent end-use quality imparted by $5 + 10$, understanding this unique origin of the wheat allele support the further exploration and evaluation of novel *Glu-D1* alleles to further improve wheat quality. This also supports the potential of unique alleles and haplotypes from the breadth of *Ae. tauschii* diversity.

Wheat grain quality remains one of the most important targets for breeders to develop superior wheat cultivars. Wild wheat relatives have been shown as a valuable resource for accessing novel genetic diversity to improve a range of wheat breeding targets including yield and disease resistance. For quality evaluation, however, the large quantity of grain needed for milling and baking and the confounding morphological characteristic needed for quality evaluation, such as suitable seed size for milling, make a direct evaluation of various end-use quality traits intractable to phenotype directly these wild relatives, including *Ae. tauschii*. In this work, we, therefore, took the first step in a reverse genetics approach in *Ae. tauschii* by identifying and characterizing variants at the important *Glu-D1* locus. This demonstrated the unique origin of the *Glu-D1* allele in wheat as well as uncovering novel allele variants and haplotypes that can now be targeted for breeding. We also established the relation of wheat alleles to those of *Ae. tauschii* and have shown that *Ae. tauschii* contains a trove of unique *Glu-D1* alleles very unlike the alleles in the current wheat germplasm. With accessible germplasm resources such as synthetic hexaploids[6], the diagnostic variants will enable marked-assisted selection of novel *Ae. tauschii* introgressions into wheat, characterization of their end-use quality, and utilization in wheat improvement.

## Methods

**Plant material**. This study included 273 *Aegilops tauschii* accessions, of which 241 were from the Wheat Genetics Resource Center (WGRC) collection at Kansas State University in Manhattan, KS, USA. Another 28 were from the National Institute for Agricultural Botany (NIAB) in Cambridge, United Kingdom. An additional 2 were from the Commonwealth Scientific and Industrial Research Organisation (CSIRO) in Canberra, Australia. The final accession, AL8/78, was obtained from the John Innes Center (JIC) in Norwich, Norfolk, England. Data regarding original collection sites for the WGRC accessions are detailed in Supplementary Data 1[5]. *Aegilops tauschii* is divided into two subspecies, spp. *tauschii* (Lineage 1) and the wheat D-genome donor spp. *strangulata* (Lineage 2). In this data set, 117 accessions were Lineage 1 and 143 Lineage 2. An additional eight accessions (five non-redundant) belonged to the newly described Lineage 3[6].

**SDS-PAGE analysis**. The sodium dodecyl sulfate-polyacrylamide gel electrophoresis (SDS-PAGE) analysis of 72 of the *Ae. tauschii* accessions was conducted at the Wheat Chemistry and Quality Laboratory at the International Maize and Wheat Improvement Center (CIMMYT) Texcoco, Mexico according to Singh et al.[35] with the following modifications. Specifically, 20 mg of wholemeal flour were mixed at 1400 rpm with 0.75 ml of 50% propanol (v/v) for 30 min at 65 °C in a Thermomixer Comfort (Eppendorf). The tubes were then centrifuged for 2 min at 10,000 rpm, and the supernatant containing the gliadins was discarded. The pellet was then mixed with 0.1 ml of a 1.5% (w/v) DTT solution in a Thermomixer for 30 min at 65 °C, 1400 rpm, and centrifuged for 2 min at 10,000 rpm. A 0.1 ml volume of a 1.4% (v/v) vinylpyridine solution was then added to the tube which was subsequently placed again in a Thermomixer for 15 min at 65 °C, 1400 rpm, and centrifuged for 5 min at 13,000 rpm. The supernatant was mixed with the same volume of sample buffer (2% SDS (w/v), 40% glycerol (w/v), and 0.02% (w/v) bromophenol blue, pH 6.8) and incubated in the Thermomixer for 5 min at 90 °C and 1400 rpm. Tubes were centrifuged for 5 min at 10,000 rpm, and 8 ml of the supernatant were used for the glutenin gel. Glutenins were separated in polyacrylamide gels (15% or 13% T) prepared using 1 M Tris buffer, pH of 8.5. Gels were run at 12.5 mA for ~19 h. Alleles were identified using the nomenclatures proposed by Payne and Lawrence[19] for bread wheat high-molecular weight glutenins and Lagudah and Halloran[24] for previously described *Ae. tauschii* high-molecular weight glutenins.

**DNA sequencing**. Whole-genome Illumina paired-end sequencing to 10× coverage for most accessions, and 30× coverage for select accessions, was obtained from TruSeq PCR-free libraries with 350 bp insert with Illumina paired-end sequencing of 150 bp according to manufactures recommendations. Sequence datasets are detailed in Gaurav et al.[6] and publicly available through NCBI sequence repositories PRJNA685125[36] and PRJNA694980[37].

**Variant calling and duplicated accessions**. Paired-end reads of the 306 *Ae. tauschii* samples were aligned to the *Ae. tauschii* 'AL8/78' genome assembly[38]. Hexaploid wheat samples were aligned to an in silico reference assembly including the hexaploid wheat A and B genomes from 'Jagger'[39] combined with the Aetv4.0 D genome using HISAT2 version 2.1.0 with default parameters[40]. Alignments were sorted and indexed using samtools v1.9[41]. Allele calls were made for all sites within the coding and 2.5 kb flanking regions of the *x* and *y* subunits of *Glu-D1* for the tauschii and wheat samples independently using bcftools version 1.9[42] 'mpileup' and 'call' commands with a minimum alignment quality of 20 (—q 20). The two resulting genomic variant call files for the *Ae. tauschii* and wheat samples were then merged with the 'merge' command of bcftools. All sites were called regardless of variant presence to avoid missing fixed variants within the two populations. Duplicated accessions were identified as sharing greater than 99.8% variant calls across their entire genome[43].

**Molecular haplotype analysis**. After merging the *Ae. tauschii* and wheat genomic variant call files, the genotype calls with mapping quality (QUAL) >40 were made on the ratios of the read depths of alternate allele (DV) to total read depths (DP) with an AWK script first presented in Milner et al., 2019[44]. Monomorphic sites with minor allele frequency less than 1% or missing calls in greater than 90% of samples were filtered out. Molecular haplotypes for *Glu-D1* were developed on the 308 resulting variant sites present in 273 of the *Ae. tauschii* accessions (listed in Supplemental Data 1) and 4 of the hexaploid wheat varieties ('Chinese Spring', 'LongReach Lancer', 'CDC Stanley', and 'CDC Landmark'). Samples sharing the same variants were considered to share the same molecular haplotype.

Genetic distances were calculated as the Euclidean distance on the A matrix of the variants in R. The A matrix was calculated with 'A.mat()' from the rrBLUP package[45] and Euclidean distances with 'dist()'. Hierarchical clustering of the genetic distances was found using hclust() and converted to a dendrogram object before plotting with the dendextend package[46].

Molecular haplotypes were designated by the subclade number of the *x* and *y* subunits together and then by the letter corresponding to the individual gene-level haplotype within. For example, molecular haplotype *x1a + y1b* represents the *a*th *x* haplotype and *b*th *y* haplotype within the subclade 1. It should be noted that letter designations across subclades have no correspondence. The *a*th *x* haplotype of subclade 1 is different than that of subclade 2. We later used the variant sites to generate a maximum likelihood tree using iqtree2 (https://github.com/iqtree/iqtree2) and visualized the unrooted version of it using iTOL[47] (Online interactive access: https://itol.embl.de/tree/7017911019641114162311859).

**Variant annotations and protein-based phylogeny**. Variants were annotated using SNPeff software[48] using standard codon tables and the AetV4 reference genome. Variant file (vcf) were further converted to nucleotide sequences using BCFtools (consensus) and translated into proteins using EMBOSS_transeq (https://www.ebi.ac.uk/Tools/st/emboss_transeq/). Protein sequences were used for phylogenetic tree construction using iqtree2 with ModelFinder[49] and 1000 bootstraps. The tree was visualized and annotated using iTOL[47].

**Colocalization of $5 + 10$ allele with Lineage 3 k-mers on 10+ bread wheat genomes**. *K*-mers were produced using Jellyfish (https://github.com/gmarcais/Jellyfish) and lineage-specific *k*-mer bins were determined as described in Gaurav et al.[6,50]. Briefly, if within a 100 kb bin, Lineage 3-specific *k*-mers counts are greater than either Lineage 1 and Lineage 2 counts by a margin (0.01% of 100k), the bin will be considered Lineage 3-specific bins. Colocalization of lineage-specific regions and $5 + 10$ alleles were plotted using ggplot2 based on lineage-specific *k*-mer bin distribution datasets and *Glu-D1* positions on the 10+ Wheat Genomes[39].

**Reporting summary**. Further information on research design is available in the Nature Research Reporting Summary linked to this article.

## Data availability

Sequencing data can be accessed through OWWC main paper (journal submission ID: NBT-PI52743) with the corresponding NCBI sequence repositories PRJNA685125 and PRJNA694980. Variant call file of the entire genome of all *Ae. tauschii* accessions can be accessed from *Zenodo* under https://doi.org/10.5281/zenodo.4317950. The variant calls for the *Glu-D1* locus are provided as Supplementary Data Set 2 with this manuscript. The population information for the *Aegilops tauschii* accessions is provided as Supplementary Data Set 1. *Ae. tauschii* lineage-specific *k*-mer counts in wheat genome assemblies can be accessed from *Zenodo* under https://doi.org/10.5281/zenodo.4474428.

## Code availability

Code for sequence alignments and variant calling is available at https://github.com/wheatgenetics/owwc/tree/master/variant_call_and_redundancy. R code used to generate figures is available at https://github.com/emilydelorean/Glu-D1_Aegilops-tauschii. Code for SNP effects (SNPeff) and iqtree and *k*-mer co-distributions and related analysis can be found here: https://github.com/umngao/GluD1_commsBio2021.

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

## Acknowledgements

E.D. was supported through Monsanto's Beachell-Borlaug International Scholars Program. This material is based upon work supported by the National Science Foundation under Award No. 1822162 "Phase II IUCRC at Kansas State University Center for Wheat Genetic Resources WGRC" and Award No. 1339389 "GPF-PG: Genome Structure and Diversity of Wheat and Its Wild Relatives". B.B.H.W. was supported by the UK

Biotechnology and Biological Sciences Research Council Designing Future Wheat Institute Strategic Programme BB/P016855/1. Any opinions, findings, and conclusions, or recommendations expressed in this material are those of the author(s) and do not necessarily reflect the views of the National Science Foundation.

## Author contributions

E.D. analyzed data, interpreted results, designated haplotypes, composed figures, and wrote the manuscript. L.G. contributed to sequence alignment, variant calling and annotation, and explored GATK method of variant calling to identify potential cysteine changes, Fig. 1 generation, and manuscript editing. M.I.B. contributed to SDS-PAGE analysis and manuscript editing. J.F.C.L. contributed to SDS-PAGE analysis. B.B.H.W. contributed to experimental conceptualization and design, and manuscript editing. OWWC provided *Ae. tauschii* accessions, geographic sample information and DNA sequence data. J.P. conceptualized experiments, experimental design and analysis, wrote and edited the manuscript.

## Competing interests

The authors declare no competing interests.

## Additional information

# Open Wild Wheat Consortium

Ali Mehrabi[4], Alison Bentley[2], Amir Sharon[5], Beat Keller[6], Brande Wulff[3], Brian Steffenson[7], Burkhard Steuernagel[8], Carolina Paola Sansaloni[2], Deng-Cai Liu[9], Evans Lagudah[10], Firuza Nasyrova[11], Gina Brown-Guedira[12], Hanan Sela[5], Jan Dvorak[13], Jesse Poland[1], Klaus Mayer[14], Ksenia Krasileva[15], Kumar Gaurav[8], Long Mao[16], Mario Caccamo[17], Martin Mascher[18], Mingcheng Luo[13], Parveen Chhuneja[19], Rob Davey[20], Justin Faris[21], Steven Xu[21], Paul Nicholson[8], Noam Chayut[8], Mike Ambrose[8], Nidhi Rawat[22] & Vijay K. Tiwari[22]

[4]Department of Agronomy and Plant Breeding, Ilam University, Ilam, Iran. [5]School of Plant Sciences and Food Security, Institute for Cereal Crops Improvement, Tel Aviv University, Tel Aviv, Israel. [6]Department of Plant and Microbial Biology, University of Zurich, Zürich, Switzerland. [7]Department of Plant Pathology, University of Minnesota, Saint Paul, MN, USA. [8]John Innes Centre, Norwich Research Park, Norwich, UK. [9]Triticeae Research Institute, Sichuan Agricultural University, Chengdu, China. [10]Commonwealth Scientific and Industrial Research Organization (CSIRO), Agriculture and Food, Canberra, ACT, Australia. [11]Institute of Botany, Plant Physiology and Genetics, Tajik National Academy of Sciences, Dushanbe, Tajikistan. [12]USDA-ARS, North Carolina State University, Raleigh, NC, USA. [13]Department of Plant Sciences, University of California, Davis, CA, USA. [14]Plant Genome and Systems Biology, Helmholtz Center Munich, Neuherberg, Germany. [15]Department of Plant and Microbial Biology, University of California, Berkeley, CA, USA. [16]Institute of Crop Science, Chinese Academy of Agricultural Sciences, Beijing, China. [17]National Institute of Agricultural Botany, Cambridge, UK. [18]Leibniz-Institute of Plant Genetics and Crop Plant Research (IPK) Gatersleben, Seeland, Germany. [19]School of Agricultural Biotechnology, Punjab Agricultural University, Ludhiana, India. [20]Earlham Institute, Norwich Research Park, Norwich, UK. [21]USDA-ARS, Cereal Crops Research Unit, Fargo, ND, USA. [22]Department of Plant Science and Landscape Architecture, University of Maryland, College Park, MD, USA.

