## [Transparent Peer Review File · Communications Biology]

Reviewers' comments:

Reviewer #1 (Remarks to the Author):

The major claim of the paper is that it demonstrates a unique origin of the gene conferring superior wheat breadmaking quality. Wheat quality is affected by the alleles of storage proteins called HMW glutenins. There are six HMW glutenins in wheat that occur as tandem pairs of genes on chromosomes 1A, 1B and 1D. The greatest impact on end-use quality is conferred by a gene pair on Chromosome 1D called 5+10. In wheat germplasm globally, there is only one other major glutenin D-genome allele (conferring inferior bread making quality) called Glu-D1 allele 2+12. One (or possibly two genotypes of) *Ae. tauschii* was the donor of the D genome of extant hexaploid wheat. The MS describes a survey of the allelic diversity of genes encoding glutenins in a subset of *Ae. tauschii* germplasm. The superior quality Glu-D1 allele 5+10 was assumed to derive from the *Ae. tauschii* D-genome donor.

Major comments

1. The MS states (lines 193-195):

'Although we found an *Ae. tauschii* haplotype identical to the wheat 2+12 allele haplotype, the exact wheat 5+10 haplotype was not detected in this panel, although a very closely related Lineage 3 haplotype was found... (line 201) with only eight variant differences.'

Thus, the origin of the 5+10 allele in bread wheat remains unknown. The authors suggest that further work will be required to discover its origin (Line 202). Given this, the title of the paper is confusing. What is meant by 'unique origin'? I feel that additional experiments to discover whether 5+10 does originate in *Ae. tauschii* and if so, in which line, would strengthen the case for publication in *Communications Biology*.

2. In the Conclusion (Line 259-261) it is stated, with reference to the 5+10 allele, that 'The early expression and greater transcription of this allele is also greater than that of the other Glu-D1 alleles, in particular 2+12 (ref 14). It is unclear which of these characteristics, or the combination of the two, lend 5+10 the superior quality characteristics.'

If the transcription (pattern and/or extent) of 5+10 differs from that of 2+12 in bread wheat, might this be more important than the protein sequence differences? This needs some thought because if true, it completely undermines the study of allelic diversity in *Ae. tauschii*.

3. The results show that, as expected (from the domestication bottleneck) there is a much greater diversity of Glu-D1 genes in *Ae. tauschii* than in hexaploid bread wheat. The gene sequences are compared and the SDS-PAGE mobilities of these haplotypes are determined and compared with those of the two major alleles of Glu-D1 in bread wheat. This shows that sets of haplotypes share the same SDS-mobility. Since the mobility is determined by the protein (not DNA) sequences, it would be interesting to compare these. This might shed light on the differences between 5+10 and 2+12 that are known to confer differing breadmaking quality. Furthermore, I suppose that many of the variations in *Ae. tauschii* Glu-D1 DNA sequence observed are synonymous? If so, defining a subset of haplotypes with differing protein sequences might prove useful to the community for further evaluation.

Minor points

1. The three lineages in *Ae. tauschii* are mentioned repeatedly in the MS but are not defined. Since there are three clades of haplotypes this is somewhat confusing for the uninitiated. Although the lineages are defined in the accompanying paper, I feel it would be useful to define them in the introduction to this MS also.

2. Fig. 1. The colours of the text do not match the key to lineage (in my version).

3. Fig. 2. The legend does not match the key- which is correct?

Reviewer #2 (Remarks to the Author):

Congratulations to Delorean et al for this clearly presented, typo-free, and excellently written

manuscript. I learned few new things from your work. My only comment is that Figure 1 is a bit too condensed with information. The information could/should also be shown as supplemental just so readers can have the information. It would also help to make the labels (within clades, and also axes) bigger for better readability.

Reviewers' comments:

We appreciate the helpful comments from the respective reviewers which have assisted us with improving the manuscript. We have carefully addressed each of these comments below, adding new analysis and discussion to the manuscript as requested.

Reviewer #1 (Remarks to the Author):

The major claim of the paper is that it demonstrates a unique origin of the gene conferring superior wheat breadmaking quality. Wheat quality is affected by the alleles of storage proteins called HMW glutenins. There are six HMW glutenins in wheat that occur as tandem pairs of genes on chromosomes 1A, 1B and 1D. The greatest impact on end-use quality is conferred by a gene pair on Chromosome 1D called 5+10. In wheat germplasm globally, there is only one other major glutenin D-genome allele (conferring inferior bread making quality) called Glu-D1 allele 2+12. One (or possibly two genotypes of) Ae. tauschii was the donor of the D genome of extant hexaploid wheat. The MS describes a survey of the allelic diversity of genes encoding glutenins in a subset of Ae. tauschii germplasm. The superior quality Glu-D1 allele 5+10 was assumed to derive from the Ae. tauschii D-genome donor.

Major comments

1. *The MS states (lines 193-195):
'Although we found an Ae. tauschii haplotype identical to the wheat 2+12 allele haplotype, the exact wheat 5+10 haplotype was not detected in this panel, although a very closely related Lineage 3 haplotype was found.... (line 201) with only eight variant differences.'
Thus, the origin of the 5+10 allele in bread wheat remains unknown. The authors suggest that further work will be required to discover its origin (Line 202). Given this, the title of the paper is confusing. What is meant by 'unique origin'? I feel that additional experiments to discover whether 5+10 does originate in Ae. tauschii and if so, in which line, would strengthen the case for publication in Communications Biology.*

We appreciate this comment and have added several analyses to further support the L3 origin of the 5+10 allele found in wheat. In addition, we have revised the respective discussion sections to more clearly state these findings and the respective conclusion.

- 1) In this study, there are a limited number of L3 accession surveyed and we therefore expect limited opportunity to identify the exact accession that contributed the original 5+10 allele to hexaploid wheat many millennia ago. Further, to this we also expect some sequence divergence since the original hybridization. We are therefore, not claiming to have identified the exact accession that was the L3 donor to wheat, rather just to say that this well-known 5+10 allele for quality originated from Lineage 3.
- 2) To look at any non-synonymous / potentially functional variants (addressing comments below) we have added analysis of translated protein sequence in the manuscript. From this analysis, we identified L3 accessions that were like most similar amino acid sequences to the superior wheat Glu-D1 5+10 allele. This evidence further supports that the 5+10 allele is most closely related to L3 accessions is additional support that that it originated from L3.
- 3) For additional analysis supporting the L3 origin of the 5+10 allele, we have compared the genome-wide assignment for lineage specific k-mers (data from capstone paper, Guarav et al., Nature Biotech, in review). Our further analysis of this data, overlaid with the

positions and allele status of the wheat Glu-D1 alleles, shows that the L3 region based on lineage specific *k*-mer on chromosome 1D coincides with the localization of Glu-D1 5+10 allele in the assembled 10+ wheat genomes. This is additional supporting evidence that Glu-D1 5+10 is derived from L3. We have added this information to the results and highlight the findings and observation in the new Figure 1b.

2. *In the Conclusion (Line 259-261) it is stated, with reference to the 5+10 allele, that 'The early expression and greater transcription of this allele is also greater than that of the other Glu-D1 alleles, in particular 2+12 (ref 14). It is unclear which of these characteristics, or the combination of the two, lend 5+10 the superior quality characteristics.' If the transcription (pattern and/or extent) of 5+10 differs from that of 2+12 in bread wheat, might this be more important than the protein sequence differences? This needs some thought because if true, it completely undermines the study of allelic diversity in Ae. tauschii.*

We appreciate that the reviewer pointed out the lack of clarity here. We have updated the discussion to better explain (lines 314-341). Current evidence suggests that number and location of cysteine residues play a part, as well as the rate and timing of protein accumulation. Other factors may also play a role, such as length of the central repetitive domain. The purpose of this study was to understand the *Glu-D1* diversity in *Ae. tauschii* for breeding programs utilizing introgressions from this wild relative. Our results provide context and data for markers so that the novel alleles being brought into modern wheat varieties can be differentiated from existing alleles. As the experimental lines advance, we will pair the molecular data with quality data to determine the effect these alleles have. We have updated the manuscript to clarify these points and better describe the overall approach of haplotype identification and tracking in the breeding programs as a suitable way forward. (Lines 337-341)

3. *The results show that, as expected (from the domestication bottleneck) there is a much greater diversity of Glu-D1 genes in Ae. tauschii than in hexaploid bread wheat. The gene sequences are compared and the SDS-PAGE mobilities of these haplotypes are determined and compared with those of the two major alleles of Glu-D1 in bread wheat. This shows that sets of haplotypes share the same SDS-mobility. Since the mobility is determined by the protein (not DNA) sequences, it would be interesting to compare these. This might shed light on the differences between 5+10 and 2+12 that are known to confer differing breadmaking quality. Furthermore, I suppose that many of the variations in Ae. tauschii **Glu-D1 DNA sequence observed are synonymous?** If so, defining a subset of haplotypes with differing protein sequences might prove useful to the community for further evaluation.*

We agree that defining a subset of haplotypes with differing protein sequences would be useful for the community. We have extended analysis in the paper and evaluated all of the identified variants in the *Ae. tauschii* panel. This analysis and the putative amino acid sequences are included as Supplemental Data 3.

It is important to note that this study is a diversity analysis using molecular haplotypes to tag unique alleles. We have identified synonymous / non-synonymous variants, but do not have evidence that any of our variants are functional. We are limited by not having full length gene or amino acid sequences for our samples, and with short read sequences we cannot reliably transverse the large and highly variable central repeat domain to identify copy number and

presence / absence variants in the repeat regions. We were able to accurately compare the N and C terminal domain variation. We found that most mutations were non-synonymous, as expected for a storage protein that is not under strong purifying selection. Where there were amino acid changes, we did not find any differences in the number or position of cysteine residues which would be the most promising candidates for functional variants.

We would also like to note that SDS-PAGE mobility is not correlated with protein length or superior/inferior quality. SDS-PAGE mobility is used to only determine the allele present, but as shown in this work, many alleles can be present in the same mobility pattern. Thus, different SDS-PAGE mobilities while an excellent diagnostic tool for genotyping glutenin alleles, does not in itself give any information on the functional quality of a given allele.

Minor points

1. *The three lineages in Ae. tauschii are mentioned repeatedly in the MS but are not defined. Since there are three clades of haplotypes this is somewhat confusing for the uninitiated. Although the lineages are defined in the accompanying paper, I feel it would be useful to define them in the introduction to this MS also.*

We thank the review very much for making this point and have added a paragraph in the introduction to explain the lineages (lines 56-63).

2. *Fig. 1. The colours of the text do not match the key to lineage (in my version).*

We have updated the colors to match the text. Thank you for noticing.

3. *Fig. 2. The legend does not match the key- which is correct?*

The figure has been updated so that the key and legend match. Thank you for noticing these important details.

Reviewer #2 (Remarks to the Author):

*Congratulations to Delorean et al for this clearly presented, typo-free, and excellently written manuscript. I learned few new things from your work. **My only comment is that Figure 1 is a bit too condensed with information.** The information could/should also be shown as supplemental just so readers can have the information. It would also help to make the labels (within clades, and also axes) bigger for better readability.*

We appreciate the comments from Reviewer 2. We have addressed the overly condensed Figure 1. First we have change it to an unrooted tree which more clearly demonstrates the separation of the different groups and removed the haplotype figure. In addition, we have added Figure 1b which gives further demonstration of the L3 origin of the 5+10 allele found in wheat.

REVIEWERS' COMMENTS:

Reviewer #1 (Remarks to the Author):

I feel that the authors revisions to the MS have greatly increased its clarity and I have no suggestions for further alterations. I found the MS both interesting and informative.